# Aging Interfacial Structure and Abnormal Tensile Strength of SnAg3Cu0.5/Cu Solder Joints

**DOI:** 10.3390/ma15249004

**Published:** 2022-12-16

**Authors:** Dongdong Chen, Junhu Qin, Xin Zhang, Dongcheng Liang, Hailong Bai, Jianhong Yi, Jikang Yan

**Affiliations:** 1Faculty of Materials Science and Engineering, Kunming University of Science and Technology, Kunming 650093, China; 2Tin Products Manufacturing Co., Ltd. of YTCL, Kunming 650217, China; 3R&D Center of Yunnan Tin Group (Holding) Co., Ltd., Kunming 650000, China; 4School of Engineering, Southwest Petroleum University, Nanchong 637001, China

**Keywords:** SnAg3Cu0.5 solder, interfacial reaction, tensile strength, fracture mode

## Abstract

In this study, the interfacial structure and abnormal long-term increase of tensile strength in the interfacial intermetallic compounds (IMCs) between SnAg3Cu0.5 solder and Cu substrates during isothermal aging were investigated. After reflow soldering, the IMC layer at the interface was thin and scallop-type. The interfacial layer became thicker with the increase in aging time. After 200 h of aging at 150 °C, the thickness of the interface gradually increased to 3.93 μm and the interface became smooth. Compared with the unaged Cu-Sn interface, the aged joint interface contained more Cu_3_Sn. The top of the IMC being reflown was relatively smooth, but became denser and prismatic in shape after 200 h of aging at 150 °C. The tensile strength of the joint, immediately after the reflow, reached 81.93 MPa. The tensile properties of the solder joints weakened and then strengthened as they aged. After 200 h of aging at 150 °C, the tensile strength was 83.86 MPa, which exceeded that of the unaged solder joint interface, because the fracture mode of the solder joints changed during aging.

## 1. Introduction

With increasing environmental awareness, SnAg3Cu0.5 solder is finding wide applications, owing to its overall performance which is sufficient for it to be used as a substitute for Sn-Pb solder. The rapid development of intelligent electronics and automated driving applications has increased the performance requirements of solder joints, necessitating studies on their reliability. The fracture of solder joint parts during the service period poses serious risk to life. Therefore, it is imperative to analyze the fracture modes of solder joints. The main component in the SnAg3Cu0.5 solder is Sn, whereas the remaining components are Cu and Ag. The intermetallic compound (IMC) layer plays an important role in connecting the solder alloy to the Cu substrate by nucleating between them during soldering. The IMC layer is typically thin and continuous with a scallop-type appearance. A study has reported the formation of a scallop-type Cu_6_Sn_5_ phase between a solder and Cu pad [1]. The shape and thickness of the IMC are also affected by reflow conditions, such as the solder composition, substrate (Cu) microstructure, reaction temperature and reaction time (t) [2,3,4]. Gusak et al. [5] reported a kinetic theory of flux-driven ripening by comparing the two types of ripening. Researchers have studied the diffusion modes of alloying elements during interfacial element diffusion. The theoretical model calculated the width of the scallop-type liquid channel to be 2.5 nm, which was confirmed by experiments [3]. Considering the dominant factor determining atomic diffusion, Xu et al. [6] discussed the evolution of the interfacial structure of a solder joint. Kim et al. [7] performed a kinetic analysis of the soldering reaction by determining the total volume change. Park et al. [8] developed a model and analyzed the influence of the model parameters on the morphological evolution and kinetics.

Researchers have also studied the effects of various alloying elements on the growth of IMCs. Yang et al. [9] found that the addition of Ag affected the growth orientation and coarsening behavior of interfacial Cu_6_Sn_5_ grains. Gao et al. [10] explored the effect of additives using a thermodynamic model to investigate the driving force for the growth of interfacial compounds. By adding CeO_2_ nanoparticles, Li et al. [11] determined the diffusion rate, reaction time, interfacial layer thickness and interfacial IMC grain size. With the addition of SiC nanoparticles, the fracture mechanism was characterized as a typical ductile fracture rather than a mixture of brittle and ductile fractures [12].

During the aging of solder joints, voids are generated at the interface, along with an increase in the thickness of the IMC. This deteriorates the reliability of the solder joints. Failure in solder joints is typically caused by the initiation and propagation of cracks, which leads to solder joint fracture. A study found that Kirkendall voids were formed at the interfacial layer of Cu_3_Sn/Cu under aging at 150 or 175 °C with a SnAg3Cu0.5Zn1 solder [13]. Liu et al. [14] reported a change in the fracture type from ductile (initial state) to mixed during the partial brittle fracture of SAC–Ni joints after 50 h of aging. They also analyzed the relationship between the generalized J-integral and the pseudo-crack tip opening displacement. Sun et al. [15] further proved the existence of a linear relationship between these parameters. Kuna et al. [16] used tensile tests and shear experiments to determine the necessary parameters for the material model of the Sn96Ag3Cu1 solder alloy. Lai et al. [17] discovered ductile fracture modes on the fracture surfaces of the SnCu0.7 solder.

Some researchers have predicted the reliability of solder joints by establishing corresponding models. Thambi et al. [18] built the corresponding fatigue models using standalone fatigue evaluation software and discussed the correction factors. Ding et al. [19] found that the crack paths changed from intergranular to transgranular with an increase in the loading rate. Chang et al. [20] investigated the fracture path at the interface of SnZn9Cu5Zn8 and within the SnZn9Ag0.5 solder alloy. Lai et al. [21] analyzed the crack initiation and propagation during the fracture of SnBi25 and SnBi35 solders using finite element analysis. Ye et al. [22] reported that the bend fracture energy of SnBi58 depends on the loading speed. When the joints had a high solder alloy hardness and thick IMC layer, fracture occurred in the solder alloy [23]. Li et al. [24] found that the fracture mechanism was a mixed mode, i.e., a combination of brittle and ductile fracture rather than ductile fracture alone. With an increase in the addition of Sb and isothermal storage time, Lee et al. [25] found that fracture occurred in the IMCs, but not in the solder mode or mixed mode. It was also reported that an increase in the reflow temperature increased the elastic energy release and shear strength of solder joints [26].

The aforementioned research was often carried out exclusively to study aging or fracture; however, in actual service, aging and fracture occur simultaneously. Accordingly, the change in the solder joint interface during high-temperature aging and its influence on fracture behavior requires further study.

## 2. Experimental Procedures

The SnAg3Cu0.5 (wt.%) solder was used in this study. To prepare SnAg3Cu0.5 (wt.%), high purity (>99.99%) Sn, SnCu10 (wt.%) alloy, and SnAg10 (wt.%) alloy were smelted at 350 °C for 30 min, and then cast in a steel mold to get a large solder block with a size of about 100 mm × 20 mm × 10 mm. First, the prepared solder was rolled to a thickness of 0.3 ± 0.01 mm and smeared with a flux (YT A300). The solder was placed between two Cu specimens (illustrated in Figure 1a), and the specimens were clamped and reflowed (Figure 2) to form solder joints (Figure 1b). After reflow, the samples were cooled in the air. Second, some samples were subjected to the same isothermal aging procedure for 50 h, 100 h, 150 h, and 200 h at 150 °C in an aging oven. The aging test was carried out in accordance with the standard “Test Methods for Lead-Free Solders” (JIS Z 3198). Third, a tensile test was performed to evaluate the strength of the solder joints using a tensile test machine (AG Xplus 50 kN, Shimadzu Suzhou Instruments Mfg. Co., Ltd., Suzhou, China). The tensile tests were performed at a speed of 2 mm·min^−1^ at 25 °C. For each condition, the average shear force was calculated for at least three samples. The fractured samples were placed in epoxy to permit metallographic observations. After cooling with water, the specimens were ground using SiC papers of sizes 180#, 400#, 800#, 1200#, 2000#, and 4000#, and polished using 3 μm and 0.5 μm diamond paste. The solder joints were then etched using a solution of 2% HCl, 5% HNO_3_, and 93% CH_3_OH (volume ratio).

The tensile fracture of the samples was observed using a metallographic microscope (Axio Scope A1, Carl Zeiss Microscopy GmbH, Gena, Germany) and electron microscope (Hitachi SU8010, Tokyo, Japan). The total area of the interfacial compound was obtained using the software “Image J”. The average thickness is obtained by dividing the total phase area by the image length. An energy dispersive spectrometer (EDS, Hitachi SU8010) was employed for component analysis. The fracture surfaces were analyzed using scanning electron microscopy (SEM, Hitachi SU8010). Grains oriented near their principal orientations were prepared for transmission electron microscopy (TEM) (FEI Talos F200X, Hillsboro, OR, USA) using focused-ion beam milling (FIB) (FEI Helios G4,Hillsboro, OR, USA).

## 3. Results and Discussion

### 3.1. Aging

A metallographic microscope was used to analyze the changes in interfacial morphology after reflow completion and the aging effectiveness at the macroscale. The microstructure of the SnAg3Cu0.5 solder joint with a Cu substrate is shown in Figure 3.

When the solder joint was fully reflowed, the thickness of the solder joint was 1.357 μm, and the IMC layer of the solder joint had a scalloped appearance, as shown in Figure 3a. The cross-section was uneven and partly protruded into the solder matrix. This occurs primarily because the Cu in the substrate quickly diffuses into the liquid solder during reflow, forming Cu_6_Sn_5_, and the reaction speed is relatively fast [27]. After 50 h of aging at 150 °C, the thickness of the solder joint interface increased by 32.28% to approximately 1.795 μm. At this time, the intermetallic compound–solder interface was relatively smooth. A thin layer of Cu_3_Sn existed at the interface between the IMC layer and Cu, as shown in Figure 3b.

After 100 h of aging at 150 °C, the thickness of the interface was 3.041 μm, representing a 69.42% increase (Figure 3c). At this time, there was a distinct layer of Cu_3_Sn between the interfacial Cu_6_Sn_5_ layer and Cu. After aging for 150 h, the thickness of the interface reached 3.761 μm (Figure 3d), which is an increase of 23.68%, and the interfacial morphology was similar to that after 100 h of aging. After 200 h of aging, the thickness of the interface was 3.925 μm (Figure 3e), i.e., it increased by 4.36%, and there was a marked decrease in the growth rate. The interfacial thicknesses for different aging times are shown in Figure 3f. During aging, the Cu content in the solder is relatively low, and the Cu atoms in the substrate diffuse toward the solder [5,7,10], which can promote the growth of the interfacial IMC layer.

### 3.2. Interface Layer Composition

EDS was performed to further understand the structure of the IMC/solder interface by analyzing its composition (Figure 4). The scanning results from the solder side to the Cu substrate revealed a sharp decrease in the Sn content at the interface, but an increase in the Cu content. The EDS line scanning results revealed no Ag aggregation due to low Ag content. The IMC layer at the solder joint interface was relatively thin, and the element content at the interface varied significantly [6,28,29]. In Figure 4a, the region from the solder side to the Cu substrate is divided into three areas. Area Ⅰ is the transition area between the solder and the Cu_6_Sn_5_. The Sn content in this area gradually decreased, whereas the Cu content increased. Area II comprises a Cu_6_Sn_5_ layer. The test data showed that the Sn content was relatively stable in this area. The results showed an increase in Cu content due to the limitations of equipment accuracy. Area III is the transition area between Cu_6_Sn_5_ and the Cu substrate, where the Sn content sharply decreases. The Cu content increased drastically until it penetrated the Cu substrate, after which the Cu content became stable.

Using the same analysis procedure, EDS was also performed on the solder joint interface aged for 200 h at 150 °C (Figure 4b). Because of the scanning position selection, no enriched Ag_3_Sn phase was found at the interface, and the spectral line of Ag was a straight line (Figure 4b). The range in the area was between 5% and 10%, possibly due to equipment errors. Based on the EDS results shown in Figure 4d, the thin layer corresponds to the Cu_3_Sn phase. The Cu_3_Sn layer exists only between Cu_6_Sn_5_ (Figure 4c) and the Cu substrate, and the soldering interface can be divided into the following five areas in Figure 4b. Area I is the transition area between the solder alloy and the Cu_6_Sn_5_. In this area, the Sn content decreased rapidly to approximately 60%, whereas the Cu content increased gradually. Area II is the Cu_6_Sn_5_ layer, where the Sn and Cu contents were relatively stable. Compared with Figure 4a, the thickness of the Cu_6_Sn_5_ layer increased significantly in this area. Area III represents the transition area between Cu_6_Sn_5_ and Cu_3_Sn. Area IV is the area of the Cu_3_Sn layer; the test data in this area are unstable, possibly because the Cu_3_Sn layer is thin. Area V is the transition area between the Cu_3_Sn and the Cu substrate. In this area, the Cu content increased sharply, whereas the Sn content decreased rapidly. Through EDS, the phase composition of the interfacial layer can be accurately and quantitatively analyzed [30].

### 3.3. The IMCs Morphology

Cross-sectional observations of the interface after reflow revealed that the IMC at the interface was scallop-type and extended to the interior of the solder. In this area, the IMC particle size at the interface was relatively uniform. The IMC particles were not closely connected in the part where the solder contacted the alloy compound, and there were numerous voids inside, as shown in Figure 5a,b. The scallop-type grains were relatively smooth without edges or corners. EDS analysis revealed that the interface mainly contained Cu_6_Sn_5_ particles (Figure 5c,d), and no Ag_3_Sn particles were found.

After aging for 200 h at 150 °C, the IMC layer at the interface became significantly thicker and was arranged more densely (Figure 6a). Compared with the morphology of the interface immediately after reflow completion, the smooth IMC particles at the top have completely disappeared; further, the IMC particles have a regular shape with distinct edges and corners (Figure 6b). The particle sizes of the IMCs were no longer uniform and some particles grew abnormally. Some granular Ag_3_Sn particles (Figure 6c,d) appeared on the top or in the gap of the IMC in Figure 6b, and the distribution of Ag_3_Sn was uneven, with agglomeration at the interface between the IMCs and solder.

### 3.4. Tensile Strength and Fracture Behavior of the Solder Joints

Figure 7a represents the tensile stress–strain curves of the solder joints under different aging times at 150 °C These curves were obtained by recording the overall displacements of the samples. The changes in the tensile strengths of the solder joints are shown in Figure 7b. The tensile strength of the unaged solder joint was 81.93 MPa. After aging for 50 h, the tensile strength was 69.59 MPa (a reduction of 15.06%). With an increase in the aging time to 100 h, the tensile strength was 74.62 MPa, representing an increase of 7.23% compared with that after 50 h of aging. After 150 h of aging, the tensile strength was 78.93 MPa. After aging for 200 h at 150 °C, the tensile strength was 83.86 MPa, which exceeded that of the unaged solder joint. The particles at the interface became more compact and uniform in Figure 5a and Figure 6a, which is conducive to the movement of grains during deformation. Therefore, the solder joints aged for 200 h had a higher displacement profile compared with the other time variations.

The tensile strength of the unaged solder joint reached 81.93 MPa and those of the aged joints exceeded 69 MPa. Xie et al. [31] found that the maximum tensile strength of SnAg3.9Cu0.7/Cu can reach 111.5 ± 12.0 MPa when the strain rate was 1.6 s^−1^. Zhu et al. [32] found that the average tensile strength of multi-alloyed solder joints (SnAg3.8Cu0.7Bi3Sb1.5Ni0.4In0.2Ce0.1/Cu) was approximately 160 MPa. The tensile strength of the above tests was higher than that of previous results [33]. The tensile strength of the bulk SnAg3Cu0.5 solder should be approximately 50 MPa. However, the tensile strength of the Cu/SnAg3Cu0.5/Cu joint was far greater than 50 MPa. The difference in tensile strength between the Cu/SnAg3Cu0.5/Cu joint and solder alloy was large. These questions require further in-depth research. To understand this mechanism, the fracture modes of the solder joints were analyzed.

The IMC layer at the solder interface had an uneven surface, as shown in Figure 5a,b. During the tensile test, fractures occurred in the interfacial IMC layer. Figure 8a also shows that fracture occurred at the interface between the IMC layer and solder. It was previously found that fracture occurs along the grain boundaries and cleavage planes of Cu_6_Sn_5_ [34]. Some Cu_6_Sn_5_ particles embedded deep in the solder underwent brittle fracture, and these fractured particles remained in the solder (yellow box in Figure 8b). Some cracks spread along the contact surface between the Cu_6_Sn_5_ particles and solder, i.e., some Cu_6_Sn_5_ particles peeled off, revealing the complete top-view morphology of the Cu_6_Sn_5_ particles near the fracture surface on the Cu substrate. At this point, the section on the solder side was pit-shaped, as indicated by the red circles in Figure 8a,b.

Figure 8c,d show that the fracture surface close to the Cu substrate is essentially the fracture surface of the Cu_6_Sn_5_ particles or peeled Cu_6_Sn_5_ particles, and a small amount of solder residue exists. This is because upon completion of soldering, part of the solder between the uneven Cu_6_Sn_5_ particles is pulled, cracked, and remains on the side of the fracture surface close to the solder. Depending on their size and shape, Cu_6_Sn_5_ grains undergo shear fracture at the foundation or at the center [35]. On the side of the fracture surface close to the solder, numerous Cu_6_Sn_5_ particles remained on the fracture surface after brittle fracture (Figure 8a,b). The peeling off of some Cu_6_Sn_5_ particles resulted in pits, as shown in the yellow box in Figure 8c,d.

After aging for 200 h at 150 °C, the tensile fracture mode of the solder joint exhibited a considerable change (Figure 9a–d). Compared with the unaged joints, fractures occurred in the IMC layers. Moreover, during the fracture of the IMC layer, the Cu_6_Sn_5_ grains underwent transgranular fracture as well as intergranular fracture, i.e., these fracture modes co-exist. The large Ag_3_Sn particles affect the fracture mode but do not reduce the fracture strength [36]. At the microscale, dislocation pile-up and slip mechanisms explain the origin of the fracture [37]. Thus, two fracture modes co-exist on the fracture surface, as shown in Figure 9b.

To analyze the structure of the aging interface, the interface structure was analyzed as shown in Figure 10. An uneven grain size is distributed at the interface, as shown in Figure 10a. The EDS mapping results show that this area is the solder and the Cu_6_Sn_5_ interface, as shown in Figure 10b. Cu exists mainly in the form of Cu_6_Sn_5_. Based on the analysis of the different regions in Figure 10c, different orientations of the Cu_6_Sn_5_ grains were found. Their strip axes are both [$\bar{1}$33]. From the magnified HR-TEM image shown in Figure 10d, the interplanar spacings of the (0$\bar{1}$1) and (310) planes were determined to be approximately 0.2061 nm and 0.3664 nm, respectively.

The fracture mechanism of the interfacial compound layer is similar to that of plastic deformation transfer in polycrystalline metallic materials. In metallic polycrystals, the slip system is initiated first in crystal grains with the highest orientation factor under stress, and intragranular dislocations move to the surface of the crystal grains or at the grain boundaries due to shear stress [34]. Because the dislocations cannot pass through the large-angle grain boundary, they accumulate at the grain boundary, forming dislocation blocking at the front end of the plugging group. When the number of dislocations in the plugging group reached a certain value, the plugging stress field activated the slip system in adjacent grains or formed microcracks at the grain boundaries [38]. The interfacial compound layer fractures after the slip zone, i.e., the dislocation pile-up acts on it and may cause a fracture.

The fracture position of the solder joint during the tensile test was analyzed using the schematic shown in Figure 11. During the tensile test of the newly formed solder joint, a fracture occurred at the interface between the interfacial layer and solder, and some IMC particles peeled off. A pit was retained on the solder side of the fracture surface and intact IMC particles were observed on the Cu plate side of the fracture surface. In other words, stripping occurred on the contact surface between the IMC particles and the solder (Figure 11a). During the tensile test after 200 h of aging (of the solder joint) at 150 °C, fracture occurred between the IMC layers in two modes, i.e., intergranular and transgranular; these modes existed alternately (Figure 11b).

## 4. Conclusions

The effect of aging time on the interfacial structure and abnormal fracture behavior of the interface formed between a eutectic SnAg3Cu0.5 solder and a Cu pad was studied in this work. The main conclusions are as follows.

The IMC layer at the interface after the reflow was thin and scallop-type. After 200 h of aging at 150 °C, the thickness of the interface gradually increased to reach 3.93 μm, and the interface became smooth.Analysis of the unaged solder joint interface revealed that it could be divided into three areas: the transition layer between the solder, the Cu_6_Sn_5_ layer, and the transition layer from the Cu_6_Sn_5_ layer to the Cu plate. The solder joints interface after 200 h of aging at 150 °C could be divided into five areas. Compared with the unaged solder joint interface, this interface had more Cu_3_Sn in the interfacial layer.The top of the IMC that had just been reflowed was relatively smooth, and there was a large gap between the Cu_6_Sn_5_ particles. After 200 h of aging at 150 °C, the top of the IMC layer became denser and exhibited a prismatic shape. Ag_3_Sn particles precipitated in the gap between the Cu_6_Sn_5_ particles.The tensile strength of the joint immediately after reflow reached 81.93 MPa. When the aging time was 50 h, the tensile strength decreased to 69.58 MPa. With an increase in the aging time, the tensile strength gradually increased. When the aging time was 200 h at 150 °C, the tensile strength was 83.86 MPa, exceeding that of the unaged solder joint.During the tensile fracture of the joint that had just been reflowed, fracture occurred at the interface between the IMC layer and solder, and some Cu_6_Sn_5_ particles broke, while some Cu_6_Sn_5_ particles peeled off from the solder joint interface. After 200 h of aging at 150 °C, fracture occurred in the IMC layer, where some Cu_6_Sn_5_ particles underwent intergranular fracture, and some underwent transgranular fracture.

## Figures and Tables

**Figure 1 materials-15-09004-f001:**
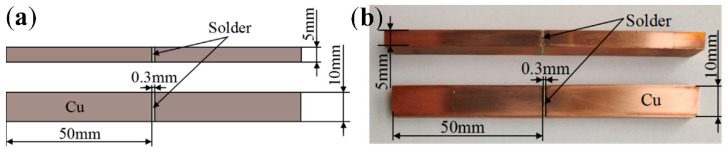
(**a**) Dimension of the test specimens and (**b**) Samples prepared after reflow.

**Figure 2 materials-15-09004-f002:**
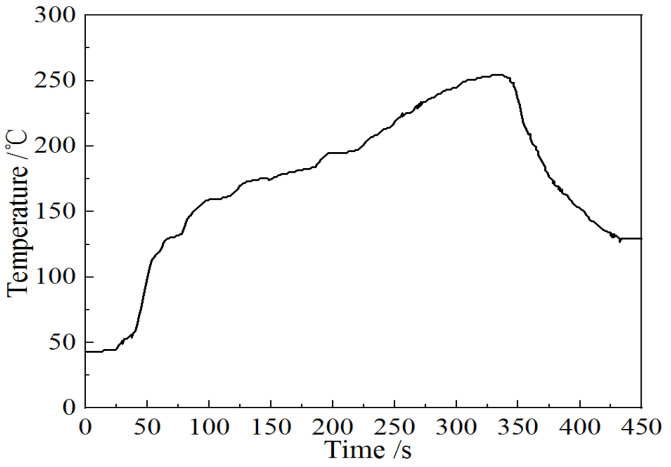
Reflow soldering curve.

**Figure 3 materials-15-09004-f003:**
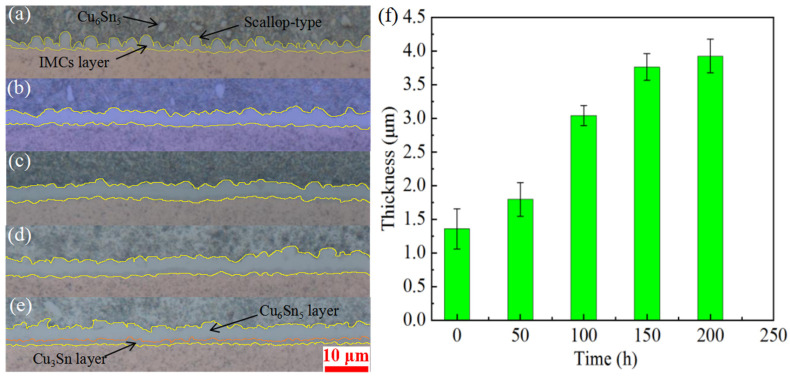
Representative micrographs showing the interface layer growth of solders at (**a**) no aging condition, t = 0 h, (**b**) 150 °C aging, t = 50 h, (**c**) 150 °C aging, t = 100 h, (**d**) 150 °C aging, t = 150 h, and (**e**) 150 °C aging, t = 200 h solder joint samples, (**f**) the thickness of the interface layer, respectively.

**Figure 4 materials-15-09004-f004:**
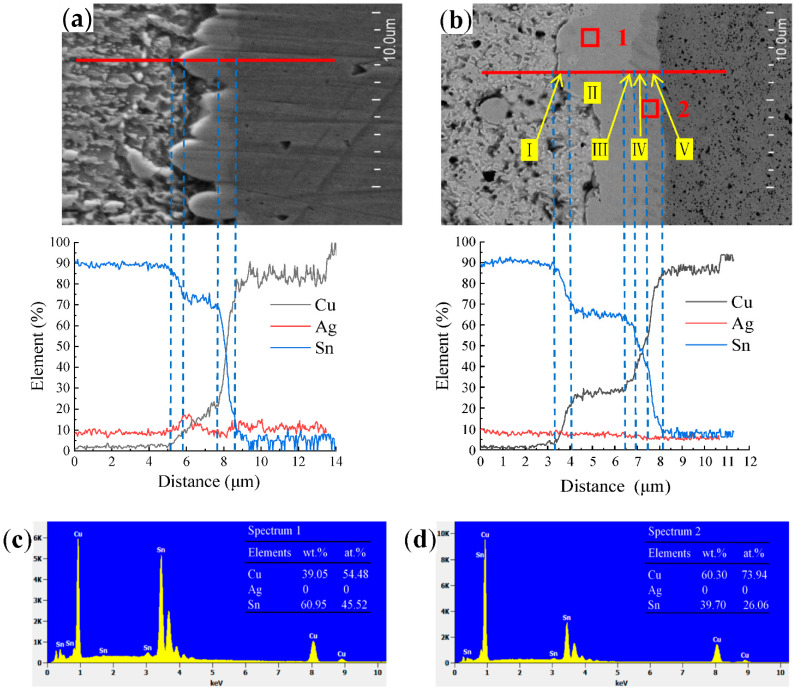
Representative SEM micrographs showing the interface layer growth and composition of solders at (**a**) no aging condition, t = 0 h, (**b**) 150 °C aging, t = 200 h solder joint samples; (**c**,**d**) shows the EDS results of interface layer in (**b**).

**Figure 5 materials-15-09004-f005:**
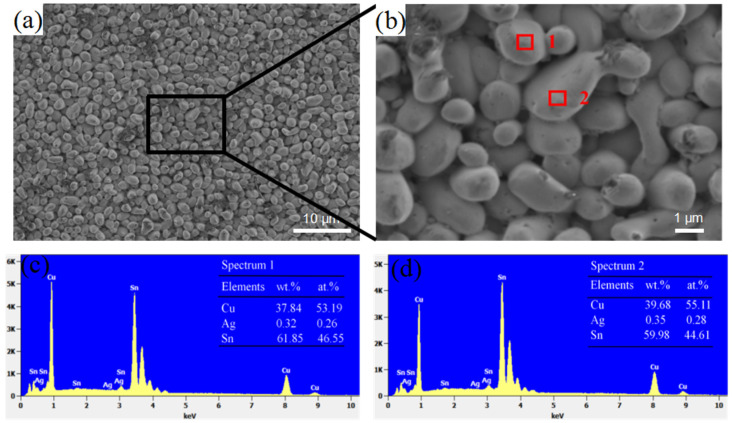
(**a**,**b**) shows the SEM images of IMCs layers formed at the solder joint interface without aging; (**c**,**d**) shows the EDS results of the rectangular 1, 2 of the interface in (**b**).

**Figure 6 materials-15-09004-f006:**
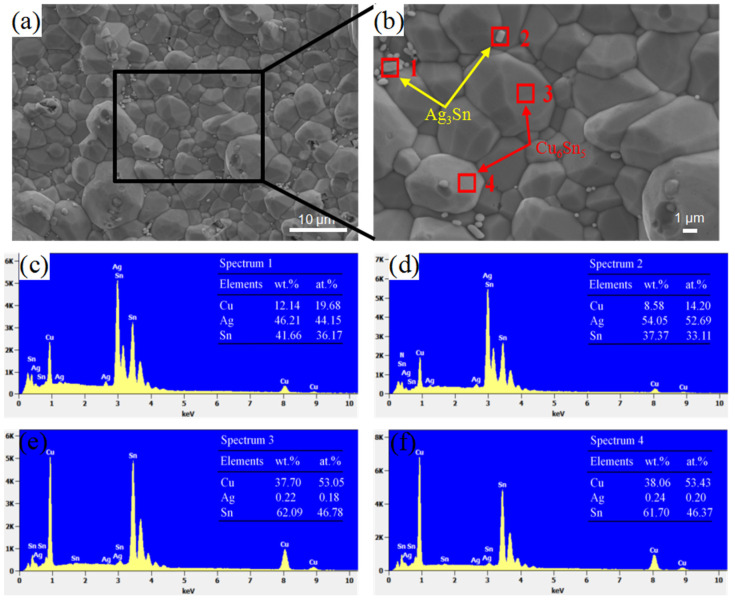
(**a**,**b**) show the representative SEM micrograph of the interface with aging at 150 °C for 200 h, (**c**–**f**) the EDS results of the interface of solder in (**b**).

**Figure 7 materials-15-09004-f007:**
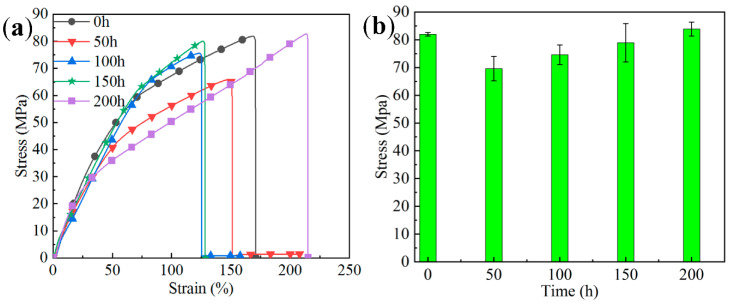
Tensile strength test, where (**a**) shows the stress–strain curves of solder joints aged at 150 °C for different times and (**b**) shows the average tensile strength.

**Figure 8 materials-15-09004-f008:**
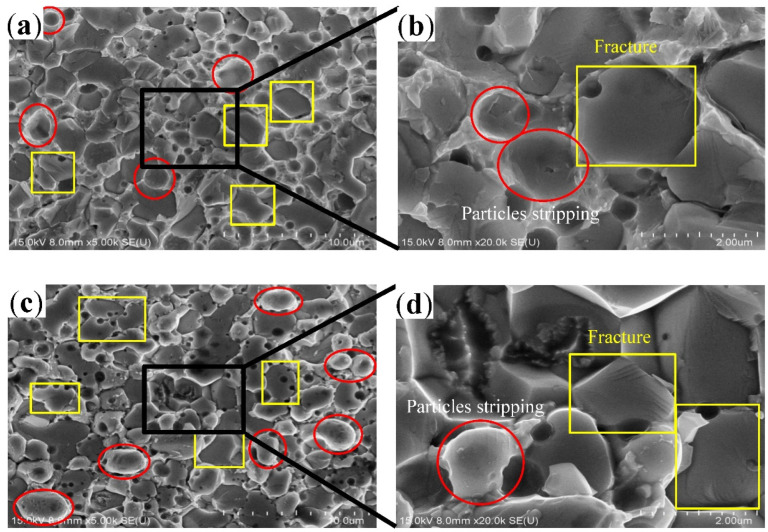
The SEM images of the fracture surfaces of solder joint, where (**a**,**b**) are the solder side, (**c**,**d**) are the Cu substrate side.

**Figure 9 materials-15-09004-f009:**
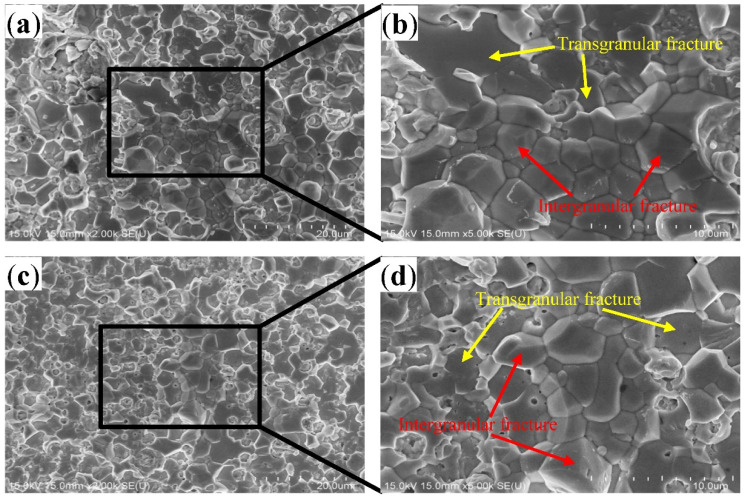
The SEM images of the fracture surfaces of solder joint aged for 200 h at 150 °C, where (**a**,**b**) are the solder side, (**c**,**d**) are the Cu substrate side.

**Figure 10 materials-15-09004-f010:**
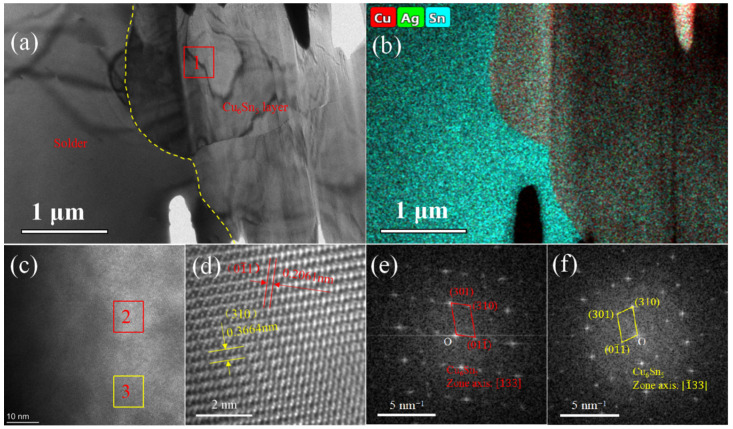
Interface microstructure analysis of the solder joint aged for 200 h at 150 °C, where (**a**) shows the TEM image, (**b**) shows the EDS-mapping results, (**c**) shows the HRTEM image of the rectangle “1” between solder and Cu_6_Sn_5_ in (**a**,**d**) a locally magnified area “3” marked in image (**c**,**e**) and (**f**) shows the SAED patterns of marked rectangle “3”, “4” in (**c**).

**Figure 11 materials-15-09004-f011:**
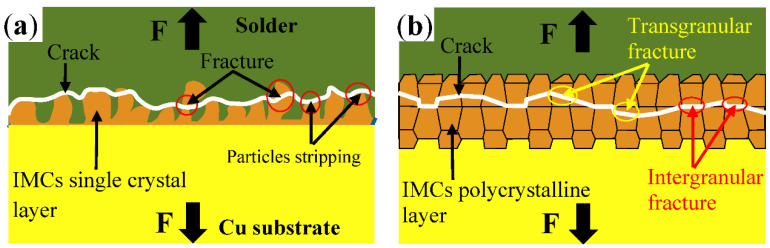
Schematic diagram of fracture location, (**a**) no aging, (**b**) aged for 200 h at 150 °C.

## Data Availability

The data presented in this study are available upon request from the corresponding author. The data are not publicly available due to the fact of technical or time limitations.

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
