# Peer review of "Aging Interfacial Structure and Abnormal Tensile Strength of SnAg3Cu0.5/Cu Solder Joints"

_materials, 2022, doi:10.3390/ma15249004_

Round 1

Reviewer 1 Report

Authors have presented an article entitled “Aging interfacial structure and abnormal long-term increase of SnAg3Cu0.5 solder joints” Though the manuscript is well written and organized but there is scope for further improving the quality of the draft before considering for publication.  

Few minor comments are listed below:

1. Please mention 1 and 2 rectangular circle caption in Figure 4b.

2. After 200 h, it got higher displacement profile compare with the other time variation. Please give the justification.

3. Authors nicely designed the results and discussion section, they should check the typos and grammatical mistakes in the revised version.  

Author Response

Dear Reviewer:

Thanks to the reviewer for the careful review of the manuscript.

The corrections in the paper and the response to the reviewer’s comments are as following:

  1. Please mention 1 and 2 rectangular circle caption in Figure 4b.

Response: According to the comments of another reviewer, the reflow soldering curve of sample preparation was added. The rectangular 1,2 have been mentioned in Figure 5.

Revised as follows:

  1. After 200 h, it got higher displacement profile compare with the other time variation. Please give the justification.

Response: The particles at the interface become more compact and uniform in Fig. 4(a) and 5(a) , which is conducive to the movement of grains in the deformation. Combined with the morphology of the fracture surface after tensile fracture,after tensile fracture of the unaged solder joint, there are only fracture grains on the solder side, without complete grain. However, After aging for 200 h, the tensile fracture mode of the solder joint exhibits a considerable change. The Cu6Sn5 grains undergo not only transgranular fracture, but also intergranular fracture. And complete grains were found at the solder side. So the solder joints aged 200h get higher displacement profile compare with the other time variation.

  1. Authors nicely designed the results and discussion section, they should check the typos and grammatical mistakes in the revised version.  

Response: Thanks for the reviewer's recognition of the manuscript, I have revised the language of the whole manuscript again, and corrected the non-standard, grammatical errors and six literatures with a little relevance.

Author Response

Dear Reviewer:

Thanks to the reviewer for the careful review of the manuscript.

The corrections in the paper and the response to the reviewer’s comments are as following:

Q1:

The shape and thickness of the IMC are also affected by reflow conditions such as the solder composition, substrate (Cu) microstructure, reaction temperature, and reaction

time(t) [2-10].

Comments:

I think the author cited too many references here [2-10].

Response: Thanks for the careful reading of the reviewer. After re-reading the 9 literatures, the 6 literatures with little relevance have been deleted. The three literatures selected were as follows:

[2] H. F. Zou, H. J. Yang, Z. F. Zhang, Coarsening mechanisms, texture evolution and size distribution of Cu6Sn5 between Cu and Sn-based solders, Mater. Chem. Phys. 131 (2011) 190-198, https://doi.org/10.1016/j.msea.2011.04.008.

[3] J.O. Suh, K.N. Tu, G.V. Lutsenko, Size distribution and morphology of Cu6Sn5 scallops in wetting reaction between molten solder and copper, Acta Mater. 56 (2008,) 1075-1083, https://doi.org/10.1016/j.actamat.2007.11.009.

[4] Y. Tian, R. Zhang, C. Hang, Relationship between morphologies and orientations of Cu6Sn5 grains in Sn3.0Ag0.5Cu solder joints on different Cu pads, Mater. Charact. 88 (2014) 58-68, https://doi.org/10.1016/j.matchar.2013.12.006.

The six references deleted were as follows:

[3] H.F. Zou, H.J. Yang, Z. F. Zhang. Morphologies, orientation relationships and evolution of Cu6Sn5 grains formed between molten Sn and Cu single crystals, Acta Mater. 56 (2008) 649-2662, https://doi.org/10.1016/j.actamat.2008.01.055.

[6] Z. Zhang, H. Cao, M. Li, Three-dimensional placement rules of Cu6Sn5 textures formed on the (111)Cu and (001)Cu surfaces using electron backscattered diffraction[J]. Mater. Des. 94 (2016) 280-285, https://doi.org/10.1016/j.matdes.2016.01.037.

[7] P.T. Lee, Y.S. Wu, C.Y. Lee, High-Speed Cu Electrodeposition and Reliability of Cu Pillar Bumps in High-Temperature Storage, J. Electrochem. Soc. 165 (2018) 647-653, https://doi.org/10.1149/2.1001813jes.

[8] P.T. Lee, Y.S. Wu, P.C. Lin, High-speed Cu electrodeposition and its solderability, Surf. Coat. Technology 320 (2016) 559-567, https://doi.org/10.1016/j.surfcoat.2016.11.016.

[9] Z. Zheng, P.C. Chiang, Y.T. Huang, Study of grain size effect of Cu metallization on interfacial microstructures of solder joints, Microelectron. Reliab. 99 (2019) 44-51, https://doi.org/10.1016/j.microrel.2019.05.018.

[10] H.L. Hsu, H.Lee, P.H. Wu, Impurity-induced unusual microstructural evolution and mechanical property in Sn/Cu solder joints, J. Mater. Sci. Mater. Electron. 29 (2018) 12842-12849, https://doi.org/10.1007/s10854-018-9403-7.

Q2:

Comments:

The author should add the coresponding sign and symbol between the two parts of Fig (a) and (b).

Response: Thanks for the opinion of the editor. By marking the corresponding parts in Figure b, the characteristics of the samples prepared after reflow are more prominent. The revised figure 1 (b) was as follows:

Q3:

Comments:

In Fig 2, please provide the method for measurement the thickness of the interface layer.

Response: Considering the irregular shape of the interface layers as shown in above figure, their thicknesses were digitally measured by using the software “Image J”. While, the average thickness of the IMC layers (x)in aged solder joints were calculated through dividing the integrated area (A) by the length of the IMC layers (L), as shown in the following equation:

This method has been used in many literatures, so it is not described in detail in the manuscript.

Q4:

Comments:

Please show to way to obtain the lower diagrams of the above figure.

Response: In order to obtain the change of element content in the interface layer, the line scanning of the interface layer before and after aging was performed, and the data obtained by line scanning were compared with the scanning position to obtain the above figure.

Q5:

Comments:

Please provide the method to determine the phase Ag3Sn as in above figure and Cu6Sn5 layer in Figure 9 (a).

Response: The solder joints were then etched with a solution of 2% HCl + 5% HNO3 + 93% CH3OH(volume ratio). The samples were deeply etched with the above etchant solution to remove the excess Sn phase so that the interface layer can be fully exposed. In order to obtain a cleaner interface layer appearance, the interface layer needs to be observed several times during the process of deeply etching until the solder is completely etched. Figure 4 and Figure 5 were obtained in this way.

Cu6Sn5 layer oriented near their principal orientations were prepared for transmission electron microscopy (TEM) (FEI Talos F200X) by focused-ion beam milling (FIB) (FEI Helios G4).

Q6:

Comments:

Could you explain the causes of intergranular and transgranular fracture in Fig. 9?

Response: It is found that there are two types of tensile fracture after aging: intergranular fracture and transgranular fracture, and polycrystalline layer also exists at Cu6Sn5 layer. Cu6Sn5 particles have different orientations at the interface, which may have different bonding strength.

Next, molecular dynamics will be used to analyze the binding strength of Cu6Sn5 with different orientations. At present, the specific fracture mechanism is still unclear.

Reviewer 3 Report

The authors present an interesting manuscript where they try to elucidate the effect of aging at 150 °C in relatively long times, the fracture behavior of Cu plate joints with SnAg3Cu0.5. However, to improve the quality of the document presented, the following is recommended:

It is recommended to show in Figure 3 the line scans semi-quantification of the other aging times, so you can establish the mechanism of growth or thickening of the interface and or the possible depletion of the adjacent copper.

Similarly to the previous point, for figure 5, it is recommended to show SEM micrographs and microanalysis of the interfaces with intermediate holding times at 150 °C to evaluate the evolution of the precipitates rich in Ag and Cu.

Figure 6 shows a significant reduction in strength and deformation at 50 hours of aging, reduction in deformation between 100 and 150 hours of aging, but at 200 hours there is a significant recovery of strength and an improvement in deformation, Can author establish if this behavior go on? The authors are recommended to make a concise explanation of this behavior based on the evolution of the precipitates.

Author Response

Dear Reviewer:

Thanks to the reviewer for the careful review of the manuscript.

The corrections in the paper and the response to the reviewer’s comments are as following:

1.It is recommended to show in Figure 3 the line scans semi-quantification of the other aging times, so you can establish the mechanism of growth or thickening of the interface and or the possible depletion of the adjacent copper.

Response: This manuscript mainly studies the fracture mode of solder joint during aging. The line scanning results in Figure 3 are used to illustrate the changes in the thickness and composition of the solders joints interface layer during aging. Combined with the fracture location, it is helpful to analyze the change of tensile fracture mode during aging. This manuscript is not intended to analyze the interface growth mechanism and diffusion mechanism in aging, so the interface line scanning data of the other aging time are not added.

2.Similarly to the previous point, for figure 5, it is recommended to show SEM micrographs and microanalysis of the interfaces with intermediate holding times at 150 °C to evaluate the evolution of the precipitates rich in Ag and Cu.

Response:

Figure 1. The SEM images of IMCs layers formed at the solder joint interface, (a) without aging, (b) aging 50h at 150 °C, (c) aging 100h at 150 °C, (d) aging 150h at 150 °C, (e) aging 200h at 150 °C.

The above figure shows the morphology of the aging different times at 150 °C. After 50 of aging at 150 °C, scallop-type particles have become relatively dense. During 50 to 150 h, the change of interface morphology is not obvious. Therefore, only photos aged for 200 h are placed in the article to compare the interface morphology without aging. And the contrast between the two is obvious. Combined with the fracture morphology after tensile fracture, it is helpful to explain the change of fracture mode.

3.Figure 6 shows a significant reduction in strength and deformation at 50 hours of aging, reduction in deformation between 100 and 150 hours of aging, but at 200 hours there is a significant recovery of strength and an improvement in deformation, Can author establish if this behavior go on? The authors are recommended to make a concise explanation of this behavior based on the evolution of the precipitates.

Response: For the samples in this tensile test, at least 3 samples were tested in each group, and the average tensile strength was taken. The tensile curve of the sample close to the average value was used as the tensile curve. Because there are still some differences between the samples under the same preparation conditions, the tensile strength can be guaranteed to be real and reliable.

Reviewer 4 Report

Dear Authors, hereby i supple my comments to "Aging interfacial structure and abnormal long-term increase of SnAg3Cu0.5 solder joints "

C1: The title is not appropriate - what is increasing?
C2: The 200h test is not considered to be too long in joint reliability tests. Can you confirm with literature, that such time period is enough for such tests? (1000+ hour tests can be found in the literature.).
C3: What is the type of the solder (commercial -> so there must be more about it, name, manufacturer, type, flux, etc.).
C4: What was the reflow method? What was the thermal profile for reflow? Without these info, the experiment is not reproducable.
C5: Results are already discussed in Chapter 2 which is not acceptable.
C6: The model of dynamic IMC behaviour is totally missing.
C7: The statistical comparison between the tensile strength results is totally missing - from this aspect, the curves are similar, but a significance analysis is needed to find out, which groups can be considered significantly different.
C8: Occasional typos can be found in the text and in the figures (e.g. Fig 7. "Partivcles".
C9: The style of the paper is inappropriate at many points - the flow of the manuscript is simply not in the general style of scientific discussion at parts. This needs a careful rewrite. Example "How about the tensile strength of bulk SnAg3Cu0.5 solder. Generally, it should be about 50 MPa. So how come the SnAg3Cu0.5 solder matrix can survive when a Cu/SnAg3Cu0.5/Cu joint is tensile tested under a stress over 50 MPa."

If the above questions can not be answered in a satisfying manner, then I will not suggest acceptance in Materials.

Author Response

Dear Reviewer:

Thanks to the reviewer for the careful review of the manuscript.

The corrections in the paper and the response to the reviewer’s comments are as following:

C1: The title is not appropriate - what is increasing?

Response:

Thanks for the reviewer's correction. The title has been modified according to the content of the manuscript. The revised title was as follows:

Aging interfacial structure and abnormal tensile strength of SnAg3Cu0.5/Cu solder joints.

C2: The 200h test is not considered to be too long in joint reliability tests. Can you confirm with literature, that such time period is enough for such tests? (1000+ hour tests can be found in the literature.).

Response: As the editor said, the aging time of 200 h is relatively short, and the literature reports the tensile test of aging solder joints for a longer time. However, the tensile strength will change greatly during aging 200 h. This manuscript mainly studies the change of tensile strength of solder joints aged for 200 h. Analyzing the change of tensile strength from the thickness, morphology and fracture mode of interface layer, So the aging time in the manuscript is 200 h.

C3: What is the type of the solder (commercial -> so there must be more about it, name, manufacturer, type, flux, etc.).

Response: 

The sample preparation process was revised as follows:

The SnAg3Cu0.5 (wt.%) solder was used in this study. To prepare SnAg3Cu0.5 (wt.%), high purity (>99.99%) Sn, SnCu10 (wt.%) alloy, and SnAg10 (wt.%) alloy were smelted at 350 °C for 30 min, and then cast in a steel mold to get a large solder block with a size of about 100 mm × 20 mm × 10 mm. First, the prepared solder was rolled to a thickness of 0.3±0.01 mm and smeared with a flux (YT A300). The solder was placed between two Cu specimens (illustrated in Fig. 1(a)), and the specimens were clamped and reflowed (Fig. 2) to form solder joints (Fig. 1(b)). After reflow, the samples were cooled in the air.

C4: What was the reflow method? What was the thermal profile for reflow? Without these info, the experiment is not reproducable.

Response:

Thanks for the comments of the reviewer, the reflow soldering curve is now provided as follows:

Figure 2. Reflow soldering curve

C5: Results are already discussed in Chapter 2 which is not acceptable.

Response: 

Thanks for the comments of the reviewer, the Chapter 2 is now provided as follows:

The SnAg3Cu0.5 (wt.%) solder was used in this study. To prepare SnAg3Cu0.5 (wt.%), high purity (>99.99%) Sn, SnCu10 (wt.%) alloy, and SnAg10 (wt.%) alloy were smelted at 350 °C for 30 min, and then cast in a steel mold to get a large solder block with a size of about 100 mm × 20 mm × 10 mm. First, the prepared solder was rolled to a thickness of 0.3±0.01 mm and smeared with a flux (YT A300). The solder was placed between two Cu specimens (illustrated in Fig. 1(a)), and the specimens were clamped and reflowed (Fig. 2) to form solder joints (Fig. 1(b)). After reflow, the samples were cooled in the air. Second, some samples were subjected to the same isothermal aging procedure for 50 h, 100 h, 150 h, and 200 h at 150 °C in an aging oven. The aging test was carried out in accordance with the standard “Test Methods for Lead-Free Solders” (JIS Z 3198). Third, a tensile test was performed to evaluate the strength of the solder joints using a tensile test machine (AG Xplus 50kN). The tensile tests were performed at a speed of 2 mm·min−1 at 25°C. For each condition, the average shear force was calculated for at least three samples. The fractured samples were placed in epoxy to permit metallographic observations. After cooling with water, the specimens were ground using SiC papers of sizes 180#, 400#, 800#, 1200#, 2000#, and 4000#, and polished using 3 μm and 0.5 μm diamond paste. The solder joints were then etched using a solution of 2% HCl, 5% HNO3, and 93% CH3OH (volume ratio).

The tensile fracture of the samples was observed using a metallographic microscope (Axio Scope A1) and electron microscope (Hitachi SU8010). The total area of the interfacial compound was obtained by the software “Image J”. The average thickness is obtained by dividing the total phase area by the image length. An Energy Dispersive Spectrometer (EDS, Hitachi SU8010) was employed for component analysis. The fracture surfaces were analyzed using scanning electron microscopy (SEM, Hitachi SU8010). Grains oriented near their principal orientations were prepared for transmission electron microscopy (TEM) (FEI Talos F200X) using focused-ion beam milling (FIB) (FEI Helios G4).

C6: The model of dynamic IMC behaviour is totally missing.

Response: The main purpose of the manuscript is to analyze the changes of tensile strength from the interface layer thickness, morphology and fracture mode, So the model of dynamic IMC behaviour was not created.

C7: The statistical comparison between the tensile strength results is totally missing - from this aspect, the curves are similar, but a significance analysis is needed to find out, which groups can be considered significantly different.

Response: For the samples in this tensile test, at least 3 samples were made in each group, and the average tensile strength was taken. The tensile curve of the sample close to the average value was used as the tensile curve. Because there are still some differences between the samples under the same preparation conditions, the tensile strength can be guaranteed to be real and reliable. The variance distribution of Tensile results is as follows:

Tensile Sample

 No aging condition (MPa)

Aging 50h  (MPa)

Aging 100h  (MPa)

Aging 150h  (MPa)

Aging 200h  (MPa)

1#

82.2493

75.528

81.7364

89.5573

86.8469

2#

81.8682

65.8728

75.5944

80.0454

80.2525

3#

81.4069

63.0779

70.3185

69.6505

82.7103

Average value  

81.84146667

68.15956667

75.8831

79.75106667

83.2699

Variance distribution (S2)

0.118630296

28.4488159

21.76974725

66.0900971

7.404261307

C8: Occasional typos can be found in the text and in the figures (e.g. Fig 7. "Partivcles".

Response: Thanks for the reviewer's criticism and correction. The whole manuscript has been checked and verified.

C9: The style of the paper is inappropriate at many points - the flow of the manuscript is simply not in the general style of scientific discussion at parts. This needs a careful rewrite. Example "How about the tensile strength of bulk SnAg3Cu0.5 solder. Generally, it should be about 50 MPa. So how come the SnAg3Cu0.5 solder matrix can survive when a Cu/SnAg3Cu0.5/Cu joint is tensile tested under a stress over 50 MPa."

Response: Thanks for the comments of the reviewer, the "How about the tensile strength of bulk SnAg3Cu0.5 solder. Generally, it should be about 50 MPa. So how come the SnAg3Cu0.5 solder matrix can survive when a Cu/SnAg3Cu0.5/Cu joint is tensile tested under a stress over 50 MPa." is now provided as follows:

The tensile strength of the bulk SnAg3Cu0.5 solder should be approximately 50 MPa. However, the tensile strength of the Cu/SnAg3Cu0.5/Cu joint was far greater than 50 MPa. The difference in tensile strength between the Cu/SnAg3Cu0.5/Cu joint and solder alloy was large. These questions require further in-depth research. To understand this mechanism, the fracture modes of the solder joints were analyzed.

The whole manuscript has been checked and verified.

Round 2

Reviewer 4 Report

Dear Authors,

thank you for your response. What i feel is that the paper is improved a lot, however there is one detail, which is not sufficient -> the statistic analysis. Can you please extend sample count? 3 sample and averaging from this count is still not acceptable from the journal's point of view. Please improve!

Author Response

Dear Reviewer:

Thank you for the careful review of the manuscript.

After reading your suggestions carefully, we transferred the samples used to test other items to test the tensile strength. Now the number of samples test results is six, and the test results can support the original experimental conclusions. The standard deviation of tensile results is as follows:

Tensile Sample

 No aging condition (MPa)

Aging 50h  (MPa)

Aging 100h  (MPa)

Aging 150h  (MPa)

Aging 200h  (MPa)

1#

82.2493

75.528

81.7364

89.5573

86.8469

2#

81.8682

65.8728

75.5944

80.0454

80.2525

3#

81.4069

63.0779

70.3185

69.6505

82.7103

4#

80.8567

73.5341

73.8657

83.6429

81.3851

5#

82.6942

68.1762

75.6638

81.6497

83.5982

6#

82.2493

67.0453

74.3398

71.4923

86.591

Average value

81.9334

69.5852

74.6231

78.9283

83.8581

Standard deviation(S)

0.6070

4.3870

3.4595

6.8948

2.4800

Thank you again for your careful review and valuable comments.
